# Frustrated edge currents in bilayers formed of $s$- and $d$-wave superconductors

Vedangi Pathak,[*] Oguzhan Can, Nirek Brahmbhatt, and Marcel Franz[†]

*Department of Physics and Astronomy & Stewart Blusson Quantum Matter Institute,*
*University of British Columbia, Vancouver BC, Canada V6T 1Z4*

(Dated: August 15, 2025)

We explore edge currents in heterostructures formed of a high-$T_c$ cuprate and a conventional $s$-wave superconductors. The resulting $d \pm is$ superconductor spontaneously breaks time reversal symmetry $\mathcal{T}$ and, remarkably, exhibits large edge currents along certain edge directions in spite of being topologically trivial. In addition we find that the edge currents are frustrated such that they appear to emerge from or flow into sample corners, seemingly violating charge conservation. Careful self-consistent solutions that guarantee charge conservation are required to understand how this frustration is resolved in physical systems. Calculations within the Ginzburg-Landau theory framework and fully self-consistent microscopic lattice models reveal intriguing patterns of current reversals depending on edge orientation, accompanied by spontaneous formation of magnetic flux patterns which can be used to detect these phenomena experimentally. Our study illuminates the interplay between time-reversal symmetry breaking and unconventional superconductivity in high-$T_c$ superconducting heterostructures, and shows that sizable edge currents are possible even in the absence of non-trivial bulk topology.

## I. INTRODUCTION

Two-dimensional (2D) heterostructures, composed of stacked layers of different 2D materials, have emerged as a fascinating area of study in condensed matter physics. These heterostructures are formed by combining materials with distinct electronic, optical, and mechanical properties, resulting in the emergence of unique phenomena not observed in individual layers. For instance, semiconductor - superconductor and superconductor - topological insulator heterostructures have been proposed as platforms to obtain Majorana modes [1, 2]. Heterostructures of superconductors with ferromagnets, anti-ferromagnets or altermagnets can also host a wide variety of topological phases. Creating heterostructures by stacking layers of atomically thin quantum materials, like graphene [3, 4] or transition metal dichalcogenides [5], with a twist between the layers has been a paradigm-shifting idea resulting in the emergence of exotic superconductivity, correlated insulator states, and quantum anomalous Hall effects.

$Bi_2Sr_2CaCu_2O_{8+x}$ (Bi-2212 or BSCCO), a well-known d-wave superconductor with a critical temperature of 90 K, has recently been exfoliated to monolayer thickness [6], opening new possibilities for d-wave superconducting heterostructures. Recent proposal [7] of a heterostructure comprising of two such cuprate monolayers stacked with a large twist angle, exhibits a $d_{x^2-y^2} + id_{xy}$ time-reversal symmetry breaking order parameter. At a twist angle nearing 45°, this $d + id'$ superconducting system is topological with bulk gap exhibiting chiral edge modes [7]. The non-trivial topology and time-reversal symmetry breaking in this system can, in principle, be probed using signatures in Josephson tunneling between the layers [8], superconducting diode effect [9], polar Kerr

effect [10], fractional and coreless vortices [11], and chiral edge currents [12]. Recent experiments [13] revealed the presence of fractional Shapiro steps, Fraunhofer patterns and superconducting diode effect in near-45° twisted samples, indicative of spontaneous $\mathcal{T}$-breaking at the interface. Proximity coupled $d + id'$ superconductor with a topological insulator has been proposed as a platform for high-temperature Majorana modes [14].

Inspired by these recent developments, in this work, we consider a heterostructure of a $d$-wave superconductor and a conventional $s$-wave superconductor. For the reasons explained below it is most convenient to study a thin layer of a conventional $s$-wave superconductor on top of a larger cuprate superconductor, such as Bi-2212, exhibiting a $d$-wave order parameter. Even though the $d$-wave SC has gapless quasiparticle excitations, the $d/s$ bilayer system can exhibit a fully gapped excitation spectrum due to its emergent $d + is$ order parameter. Indeed such a heterojunction has been proposed as a superconducting qubit platform (the "$d$-mon") as the symmetries of this system can lead to large anharmonicities as compared to conventional transmon qubits [15].

The time-reversal symmetry breaking in the $d/s$ superconducting bilayer is most easily understood from the Ginzburg-Landau (GL) free energy for the system. Taking $s$ and $d$ as smoothly varying order parameters and $f_s[s]$, $f_d[d]$ as free energy densities of the decoupled superconductors, we may write the combined free energy of the spatially homogeneous system as

$$f_0[s, d] = f_s[s] + f_d[d] + A|s|^2|d|^2 + B(s^*d + d^*s) + C(s^{*2}d^2 + d^{*2}s^2). \tag{1}$$

If both superconductors obey tetragonal symmetry, then under $C_4$ rotation, we get $s \to s$ while $d \to -d$. This implies that $B = 0$. Therefore, the leading Josephson coupling arises from the Cooper-pair cotunnelling term on the second line. Denoting the phase difference between two order parameters by $\varphi$ the resulting Josephson free

---

[*] vedangi.pathak@gmail.com
[†] franz@phas.ubc.ca

energy becomes

$$f(\varphi) = E_0 + 2C|s|^2|d|^2 \cos 2\varphi, \qquad (2)$$

where $E_0$ contains terms independent of $\varphi$. When $C > 0$ the free energy landscape exhibits two minima at $\varphi = \pm\pi/2$. The system will spontaneously break the time-reversal symmetry $\mathcal{T}$ and choose to equilibrate in one of the minima. In this $\mathcal{T}$-broken phase the bilayer can be regarded as a $d \pm is$ superconductor.

Time-reversal symmetry breaking order parameters may give rise to edge currents. In chiral superconductors such as $d_{x^2-y^2} + id_{xy}$ or $p_x + ip_y$, edge currents originate from topologically protected edge modes and can serve as a probe of the bulk topology [12, 16]. However, the presence of edge currents does not imply nontrivial topological phase in the system. In the case of the non-topological $d/s$ bilayer system, depending on the orientation of the edge, we find that $\mathcal{T}$-breaking can lead to large supercurrents at the edge. Remarkably, these edge currents are frustrated in that they appear to flow out of and into the sample corners as illustrated in Fig. 1(b).

In what follows we first deduce the presence of frustrated edge currents from a simple qualitative analysis of the GL theory near the edge. This analysis implies that edge currents will flow along a diagonal (110) edge of a $d_{x^2-y^2} + is$ superconductor (or equivalently along the straight 100 or 010 edge of a $d_{xy} + is$ superconductor). We then confirm this result using a fully self-consistent Bogoliubov-de Gennes (BdG) lattice model which allows us to estimate the current amplitude for realistic model parameters and also understand how the frustration in the presence of corners is resolved. To this end we perform fully self-consistent calculations of the BdG theory – we find that in restricted geometries only full self-consistency guarantees current conservation. Based on such solutions, we observe that the frustration associated with corners is typically resolved by supercurrents flowing into the bulk of the system to ensure that the supercurrent remains conserved. In this way we obtain often intricate patterns of edge-generated persistent currents in $s/d$ bilayers with magnitudes that can be experimentally observed.

To understand how these effects play out on longer lengthscales we then return to the GL theory. We iteratively solve multi-component GL equations on a discrete lattice to find solutions for the $d$- and $s$-wave order parameters, such that the bulk stabilizes a $d \pm is$ phase. For different geometries of this system we evaluate the supercurrent density in each of these systems and find similar edge current effects, along with spontaneously developed magnetic flux patterns. We observe that these frustrated supercurrents flow into the bulk and create a large vortex-like patterns so that the supercurrent remains conserved.

Finally, we give an estimate of the size of these supercurrents for realistic material parameters relevant to BSCCO, the leading candidate material known to exhibit the $d$-wave order parameter. For the $s$-wave material we propose the use of iron-based superconductors as they

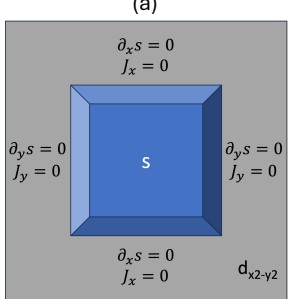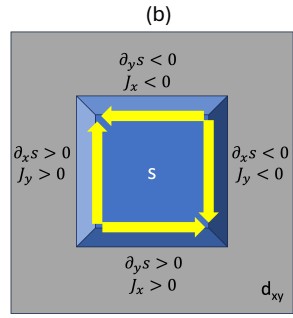

FIG. 1. The $d/s$ bilayer system comprising an $s-$wave superconducting island on a $d-$wave superconducting flake. (a) An $s$-wave island on a large $d_{x^2-y^2}$ superconducting flake. No supercurrents flow at the edges. (b) An $s$-wave island on a large $d_{xy}$ superconducting flake. At the edge of the resulting $d_{xy} \pm is$ superconductor, supercurrents flow in opposite directions on adjacent perpendicular edges, leading to supercurrent frustration at the corners. These supercurrents originate from mixed gradient terms in the GL free energy Eq. (4).

can exhibit relatively high $T_c$ and a large gap. Further, both materials have an underlying square lattice structure. We work with a long-strip geometry that is translationally invariant in one direction to estimate the edge currents. We find that the magnetic fields resulting from the edge currents are above the detection threshold of state-of-the-art SQUID microscopes.

We note here that the edge of a single, monolayer $d_{xy}$ flake can exhibit subdominant extended $s$-wave order parameter which can result in the development of the $d_{xy} + is$ order parameter at the edge of the monolayer $d_{xy}$-wave flake. In such a state, fractional vortex-antivortex pairs and phase modulations have been predicted to occur [11, 17–19]. In this work, by contrast, we focus on the edge currents that arise exclusively from the spontaneous $\mathcal{T}$-breaking in the bulk of the $d/s$ bilayer system. To ensure this, in what follows we always assume that the $d$-wave superconductor forms a large substrate and the $s$-wave layer is in the form of a finite-sized island sitting on top.

## II. FRUSTRATED EDGE CURRENTS FROM GL ANALYSIS

Let us now consider the simplest setup to illustrate the emergence of frustrated edge currents in the $d/s$ bilayer system within the GL theory Eq. (1). For this purpose, we consider a square shaped $s$-wave flake on a large $d$-wave substrate as shown in Fig. 1. In the following we will want to examine edges of the flake that are aligned with both (100) and (110) crystal directions of the $d$-wave substrate. This is most easily achieved by keeping the orientation of the flake unchanged and modeling the substrate as either $d_{x^2-y^2}$ or $d_{xy}$ superconductor as indicated in panels (a) and (b). In this setup we are effec-

tively looking at an edge of a $d_{x^2-y^2} + is$ and a $d_{xy} + is$ superconductor, respectively.

Near the edges of the flake the order parameters will not be uniform and we will need to supplement the GL free energy (1) with the relevant gradient terms. Focusing on the more interesting $d_{xy} + is$ case the full GL free energy density can be written as [20, 21]

$$f[s,d] = f_0[s,d] + \gamma_s|\vec{\Pi}s|^2 + \gamma_d|\vec{\Pi}d|^2 \qquad (3)$$
$$+ \gamma_v[(\Pi_x s)^*(\Pi_y d) + (\Pi_y s)^*(\Pi_x d) + \text{c.c.}].$$

where $\Pi = -i\nabla - e\boldsymbol{A}$ is the gauge-invariant gradient operator. The mixed gradient term on the second line is consistent with the system symmetries and underlies much of the interesting physics that we explore in this paper. Specifically, we note that it is invariant under the $C_4$ rotation which acts as $(x,y) \to (y,-x)$ and $(s,d) \to (s,-d)$, as well as the inversion and mirror symmetries.

Supercurrent density can now be deduced by varying the free energy with respect to the vector potential $\boldsymbol{A}$. At zero magnetic field we thus obtain

$$j_x \simeq ie\gamma_v(d^*\partial_y s + s^*\partial_y d) + \text{c.c.},$$
$$j_y \simeq ie\gamma_v(d^*\partial_x s + s^*\partial_x d) + \text{c.c.}, \qquad (4)$$

where we focus on contributions coming from the mixed gradient term and omit those from the conventional gradient term on the first line of Eq. (3). We notice that in Eq. (4) an $x$-derivative produces current in the $y$ direction and vice versa. Threfore, these expressions have interesting implications for edge currents in the geometry indicated in Fig. 1(b), assuming only that order parameters will be locally suppressed near the edge. Specifically, at the two horizontal edges the $y$-gradient will be positive along rising edge of the island and negative along the falling edge. Similarly, along the vertical edge, the $x$-gradient will be positive along one edge and negative along the other. This gives rise to significant supercurrents at the edge of a $d_{xy} + is$ superconductor with directions indicated by arrows in Fig. 1(b). Eq. (4) also clearly displays the importance of $\mathcal{T}$-breaking for the emergence of edge currents: for a purely real combination of $d$ and $s$ the current would identically vanish.

Remarkably, these current patterns are frustrated in that they appear to emanate from two corners of the flake and flow into the other two corners. This type of current flow would obviously be at odds with the charge conservation and motivates us to study how the physical system resolves this apparent frustration. We will address this question in the following Sections.

One can perform a similar analysis for a $d_{x^2-y^2} + is$ case shown in Fig. 1(a). The corresponding free energy density can be obtained by a straightforward 45° rotation of Eq. (3) and yields the following current densities

$$j_x \simeq ie\gamma_v(d^*\partial_x s + s^*\partial_x d) + \text{c.c.},$$
$$j_y \simeq ie\gamma_v(d^*\partial_y s + s^*\partial_y d) + \text{c.c.}, \qquad (5)$$

focusing again on the mixed-gradient terms. In this case we do not expect significant edge currents. For instance

along the horizontal edges the $x$-gradients of the order parameters will be zero leading to vanishing $j_x$ along the edge.

The main goal of the remainder of this paper is the investigation of edge currents in the $d/s$ superconducting bilayer hosting a $\mathcal{T}$-breaking $d_{xy} \pm is$ order parameter. We analyze the fate of the frustrated currents and the apparent charge non-conservation at the corners of the flake pictured in Fig. 1(b) using self-consistent BdG theory as well as the GL theory with order parameter fields computed so as to properly minimize the free energy functional. We explore different geometries and observe a number of interesting features associated with the edge current frustration present in this system.

## III. EDGE CURRENTS FROM THE MICROSCOPIC MODEL

### A. BdG Hamiltonian for the $d/s$ bilayer

We model the $s/d$ bilayer system starting with the real-space lattice Hamiltonian given by $H = H_0 + H_{\text{int}}$ with

$$H_0 = -t_s \sum_{\langle ij \rangle, \sigma s}(c_{i\sigma s}^\dagger c_{j\sigma s} + \text{h.c.}) - \mu_s \sum_{i,\sigma a} n_{i\sigma a} \qquad (6)$$

$$- t_d \sum_{\langle ij \rangle, \sigma d}(c_{i\sigma d}^\dagger c_{j\sigma d} + \text{h.c.}) - \mu_d \sum_{i,\sigma d} n_{i\sigma d} \qquad (7)$$

$$- g \sum_{i,\sigma}(c_{i\sigma s}^\dagger c_{i\sigma d} + \text{h.c.}),$$

describing the normal-state tight-binding band structure of the bilayer. Here $c_{i\sigma a}^\dagger$ creates an electron on site $i$ with spin $\sigma$ in layer $a = s, d$, $\langle ij \rangle$ denotes summation over nearest neighbor sites and $n_{i\sigma a} = c_{i\sigma a}^\dagger c_{i\sigma a}$ is the number operator. $t_a$ and $g$ denote respectively the in-plane and interplane tunneling amplitudes and $\mu$ is the chemical potential. $H_{\text{int}}$ describes attractive electron-electron interactions that give rise to $s$-wave superconductivity in one layer and $d$-wave superconductivity in the other,

$$H_{\text{int}} = -V_s \sum_{i,\sigma\sigma'} n_{i\sigma s} n_{i\sigma' s} - V_d \sum_{\langle\langle ij \rangle\rangle, \sigma\sigma'} n_{i\sigma d} n_{j\sigma' d} \qquad (8)$$

where $\langle\langle ij \rangle\rangle$ denotes summation over second-neighbor sites on the square lattice. For positive $V_a$ (and assuming decoupled layers) this form of interaction is known to produce an $s$ order in one and $d_{xy}$ order in the other layer.

Performing the standard mean-field decoupling of the interaction term (8) in the pairing channel one obtains the BdG Hamiltonian. For a uniform system with periodic boundary conditions, the Hamiltonian of the $d/s$ bilayer can be written compactly in the momentum space as $\mathcal{H} = \sum_{\boldsymbol{k}} \Psi_{\boldsymbol{k}}^\dagger h_{\boldsymbol{k}} \Psi_{\boldsymbol{k}}$ where $\Psi_{\boldsymbol{k}} =$

$(c_{\boldsymbol{k}\uparrow s}, c^\dagger_{-\boldsymbol{k}\downarrow s}, c_{\boldsymbol{k}\uparrow d}, c^\dagger_{-\boldsymbol{k}\downarrow d})^T$ and

$$h_{\boldsymbol{k}} = \begin{pmatrix} \xi_{\boldsymbol{k},s} & \Delta_{\boldsymbol{k},s} & g & 0 \\ \Delta^*_{\boldsymbol{k},s} & -\xi_{\boldsymbol{k},s} & 0 & -g \\ g & 0 & \xi_{\boldsymbol{k},d} & \Delta_{\boldsymbol{k},d} \\ 0 & -g & \Delta^*_{\boldsymbol{k},d} & -\xi_{\boldsymbol{k},d} \end{pmatrix}. \quad (9)$$

The normal state for each monolayer has a dispersion $\xi_{\boldsymbol{k},a} = -2t_a(\cos k_x + \cos k_y) - \mu_a$, where $a = s, d$. The superconducting order parameters assume the form

$$\Delta_{\boldsymbol{k}s} = \Delta_s,$$
$$\Delta_{\boldsymbol{k}d} = \Delta_d(2\sin k_x \sin k_y), \quad (10)$$

and their amplitudes are determined self-consistently from the gap equation,

$$\Delta_a = 2V_a \sum_{\boldsymbol{k},\alpha} \frac{\partial E_{\boldsymbol{k}\alpha}}{\partial \Delta^*_a} \tanh\left(\frac{\beta E_{\boldsymbol{k}\alpha}}{2}\right)$$
$$= 2V_a \sum_{\boldsymbol{k},\alpha} \langle \boldsymbol{k}\alpha | \frac{\partial h_{\boldsymbol{k}}}{\partial \Delta^*_a} | \boldsymbol{k}\alpha \rangle \tanh\left(\frac{\beta E_{\boldsymbol{k}\alpha}}{2}\right), \quad (11)$$

where $\beta = 1/k_B T$ is the inverse temperature, $|\boldsymbol{k}\alpha\rangle$ are eigenstates of $h_{\boldsymbol{k}}$ belonging to positive eigenvalues $E_{\boldsymbol{k}\alpha}$ ($\alpha = 1, 2$), and the second line is convenient in numerical computations.

We want to calculate and characterize supercurrents due to time-reversal symmetry breaking in this system. On a lattice, the current flowing along a bond between sites $i$ and $j$ in the tight binding model Eq. (6) is most easily obtained by performing the Peierls substitution $t_{ij} \to t_{ij} \exp\left(ieA_{ij}/\hbar\right)$ where $A_{ij}$ is the vector potential integrated along the bond. The current operator then follows from

$$j_{ij,a} = \frac{\partial H}{\partial A_{ij}}\bigg|_{A=0} = -\frac{et}{\hbar} \sum_\sigma \left(ic^\dagger_{i\sigma a}c_{j\sigma a} + \text{h.c.}\right), \quad (12)$$

and the last expression holds when sites $i$ and $j$ are nearest neighbors.

### B. Supercurrents in the long-strip geometry

As we expect the supercurrents to be concentrated near the edges, we first consider a long strip of the $d/s$ bilayer. We assume translational invariance along the $x$ direction and impose periodic boundary conditions in this 'long' direction. The strip has a finite width of $N_y$ unit cells along the $y$ direction and boundary conditions discussed below. Accordingly, we convert the mean-field BdG theory following from the Hamiltonian Eqs. (6,8) to the strip geometry by taking a partial Fourier transform along $x$ using

$$c_{y\sigma a}(k) = \sum_x \frac{e^{ikx}}{\sqrt{N_x}} c_{(x,y)\sigma a}. \quad (13)$$

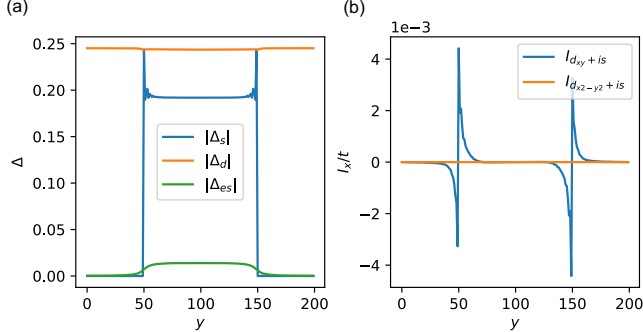

FIG. 2. Order parameters and supercurrents on a long-strip geometry of the $d/s$ bilayer system. (a) Spatial profile of the superconducting order parameters of the step edge configuration in the long strip geometry as a function of $y$. The self-consistent solutions clearly show the near-uniform $d_{xy}$ order parameter, $|\Delta_d|$, and the step edges in the $s$-wave order parameter, $|\Delta_s|$. The magnitude of the extended $s$-wave, $\Delta_{es}$, does not affect the supercurrent behavior. (b) Spatial profile of the supercurrent flowing in $x$ as a function of $y$ for the step edge configuration. Supercurrents due to the $d_{xy}+is$ order parameter are localized at the step edges. Supercurrents due to the $d_{x^2-y^2}+is$ order parameter are zero. Parameters: $t_s = t_d = 1$, $\mu_s = -1.2t_s$, $\mu_d = -1.2t_d$, $g = 0.1t_d$, $T = 0.01$, $V_{dxy} = 1$, $V_s = 0.5$.

Here $\boldsymbol{R} = (x, y)$ is used to denote lattice site position and $N_x$ denotes the number of unit cells along the periodic direction. We numerically diagonalize the resulting BdG Hamiltonian matrix, and from its eigenvectors and eigenvalues, we compute the order parameters and currents that flow along the $x$-direction, varying spatially along the $y$-direction.

Our main goal is to evaluate supercurrents generated by the $d + is$ order parameter emerging from the bilayer system. However, as mentioned in Introduction, a pair-breaking edge of the $d_{xy}$ layer can host a competing extended $s$-wave order parameter, $\Delta_{es}(2\cos k_x \cos k_y)$. This subdominant extended $s$-wave is significant and may lead to the development of a $d + is$ order parameter localized at the edges [12]. Therefore, the pair-breaking edge of the $d_{xy}$ layer can host large supercurrents that are unrelated to the effect we wish to explore here. To circumvent the issue, we assume periodic boundary conditions in the $y$ direction for $d$ electrons and introduce step edges in the $s$-wave layer. These step edges will then host edge currents that arise solely from the $d + is$ order parameter associated with the bulk of the bilayer system. Physically, this setup corresponds to a long strip of an $s$-wave superconductor placed on a long cylinder made of a $d$-wave superconductor.

We model this step edge configuration in the long strip geometry with a width of $N_y$ and maintain periodic boundary conditions along the $x$-direction in both layers. In the $s$ layer, we impose a large potential barrier across half the strip, precluding the emergence of $s$ order parameter in the region with the barrier. We model this

barrier as

$$V_{\text{barrier}} = \begin{cases} V_\infty, & \text{if } 0 \leq y < 0.25N_y \\ 0, & \text{if } 0.25N_y \leq y < 0.75N_y \\ V_\infty, & \text{if } 0.75N_y \leq y < N_y \end{cases} \quad (14)$$

with $V_\infty \gg t$.

The self-consistent solution yields a $d_{xy} + is$ order parameter in the bulk, as shown in Fig. 2(a). We obtain step edges of the $s$ order parameter in one layer at $y = 0.25N_y$ and $y = 0.75N_y$ while generating a nearly uniform $d_{xy}$ order parameter in the other layer without any edges, as depicted in Fig. 2(a).

We now analyze the edge currents in this configuration and find that supercurrents are localized near the step edges and decay into the bulk of the $d + is$ region between the two edges, Fig. 2(b). The small extended $s$-wave present in the bulk does not contribute to the supercurrent, as indicated by the absence of supercurrents in that region.

We note here that we can generate a $d_{x^2-y^2}$ order parameter given by

$$\Delta_{\boldsymbol{k}d} = \Delta_d(\cos k_x - \cos k_y) \quad (15)$$

if we sum over nearest-neighbor sites instead of next-nearest-neighbor sites in Eq. (8). Modelling the $d/s$ bilayer with the $d_{x^2-y^2} + is$ order parameter in the same step-edge configuration does not produce any supercurrents as shown in Fig. 2. In other words, the (010) edge of the $d_{xy} + is$ superconductor exhibits edge currents whereas that of $d_{x^2-y^2} + is$ superconductor does not. We already argued that this would be the case based on the GL theory. Alternately, as we explain below, we can understand this from the spectral functions of the two superconductors.

For a long strip geometry where crystal momentum $k$ along $x$ is a good quantum number, starting from Eq. (12) it is possible to derive a mixed representation expression for the current $J_{\hat{x}}(y)$ along a horizontal bond at distance $y$ from the edge in terms of the spectral function [12],

$$J_{\hat{x}}(y) = \frac{2et}{\hbar} \int_{-\pi}^{\pi} \frac{dk}{2\pi} \sin k \int_{-\infty}^{\infty} \frac{d\omega}{2\pi} \frac{\text{Tr}[A_k(y,\omega)]}{1 + e^{\beta\omega}}. \quad (16)$$

Here the trace extends over layer and BdG indices while

$$A_k(y,\omega) = -2\text{Im}[\omega + i\delta - \mathcal{H}(k)]_{yy}^{-1} \quad (17)$$

is the spectral function evaluated at distance $y$ from the edge. $\mathcal{H}(k)$ is the $4N_y \times 4N_y$ matrix Hamiltonian describing the strip and $\delta$ denotes a positive infinitesimal. Subscript $yy$ indicates that a diagonal $4 \times 4$ block of the matrix at spatial position $y$ is to be taken. Individual elements of each such $4 \times 4$ block can be thought of as position- and layer-resolved normal and anomalous components of the Gorkov Green's function $\hat{G}_k(y,y;\omega)$.

We show the spectral function $A_k(y,\omega)$ for a strip of $d_{x^2-y^2} + is$ and $d_{xy} + is$ superconductor in Fig. 3. Since

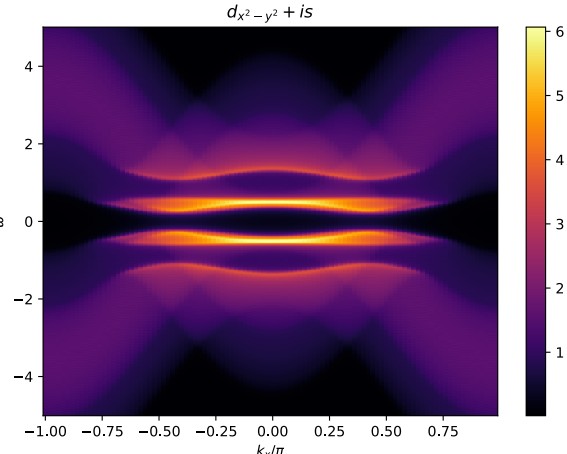

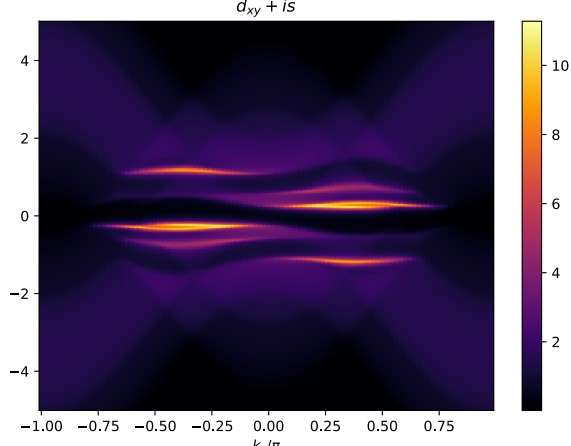

FIG. 3. Spectral functions Eq. (17) for a $d_{x^2-y^2} + is$ (top row) and $d_{xy} + is$ superconductor (bottom row) on a long strip of width $N_y = 100$. Parameters: $t_s = t_d = 1.2$, $|\Delta_d| = |\Delta_s| = 0.5$, $\mu_s = \mu_d = -1$, $g = 0.8$, $\beta = 100$.

both are non-topological with a Chern number 0, we do not see any protected edge modes. Combined with Eq. (16), these spectral function plots highlight key distinctions between the $d_{x^2-y^2} + is$ and $d_{xy} + is$ cases. Notably, due to the Fermi function, only occupied states (corresponding to the $\omega < 0$ part of the spectral function) contribute to the current in the $T \to 0$ limit. In the $d_{xy} + is$ case, spectral modes contribute primarily for $k < 0$, and this contribution is uniformly weighted by the $\sin k$ term in Eq. (16). Thus, integration over $k$ in this case results in a significant contribution to the current. In contrast, for the $d_{x^2-y^2} + is$ case, the occupied modes appear symmetrically for both positive and negative $k$, leading to cancellation in the $k$-integral when weighted by the odd $\sin k$ function.

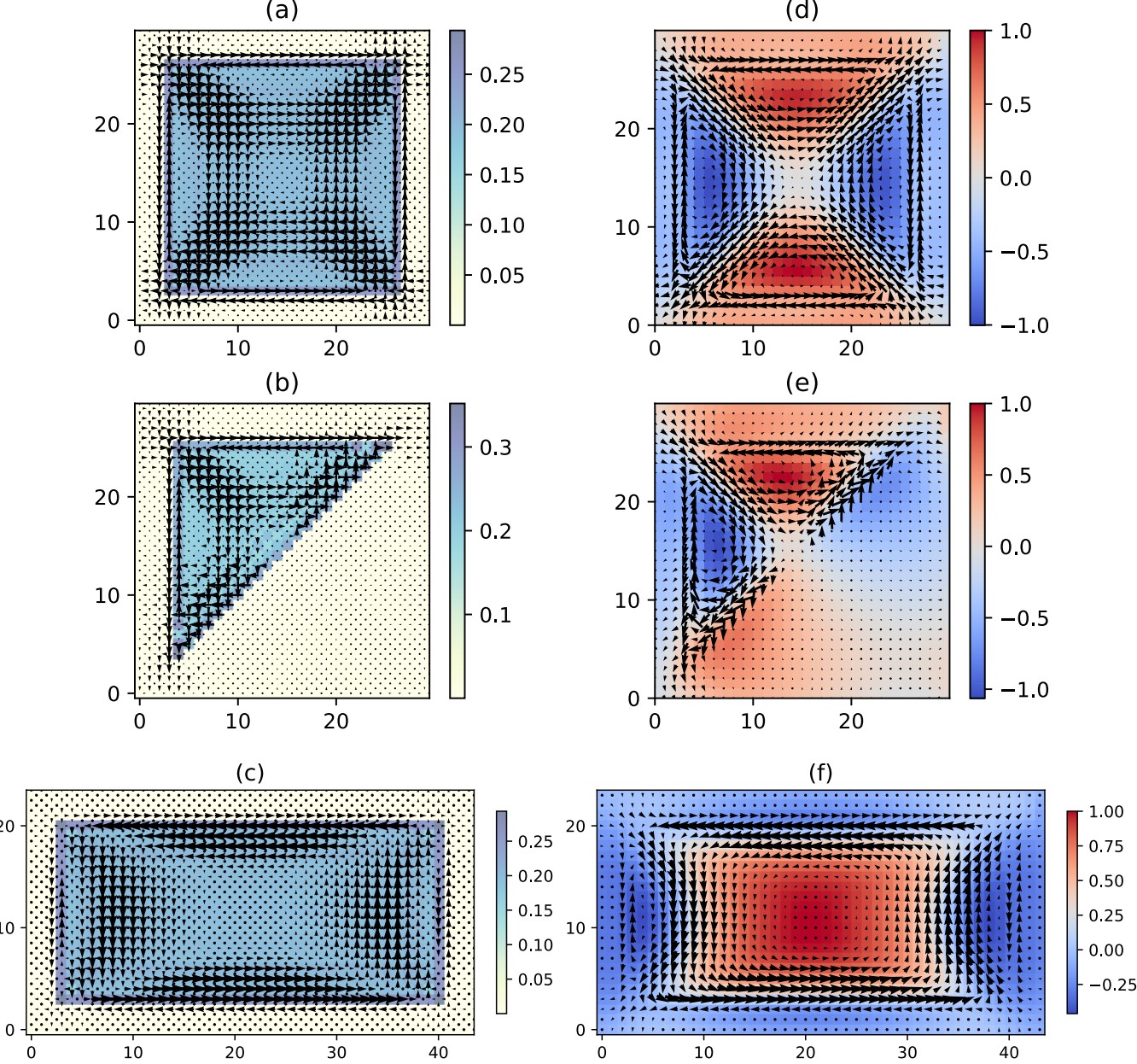

FIG. 4. Supercurrents from microscopic model for the $d/s$ bilayer system with $s-$wave islands. Panels (a),(b) and (c) depict the bond currents along the horizontal and vertical bonds for a square, a triangular, and a rectangular $s-$wave island respectively. In these configurations, the bulk stabilized a $\Delta_{d_{xy}} = 0.3$ and $\Delta_s = 0.2$. The background color in panels(a-c) depict the magnitude of the $s-$wave order parameter in the system. Panels (d), (e), and (f) display the bond currents as vectors in the $x-y$ plane, while the background color represents the normalized perpendicular magnetic field generated by the bond currents. Parameters: $t_s = t_d = 1$, $\mu_s = \mu_d = -1.2$, $g = 0.1$.

## C. Supercurrents in the island geometry

Supercurrents generated by a $d_{xy} + is$ order parameter are localized near the edges of a long strip of the $s/d$ bilayer, specifically at the step edge of the s-wave superconducting layer in the configuration we are considering. They flow in opposite directions for opposite edges simi-

lar to a chiral superconductor. However, we know that a $d + is$ order parameter is not chiral.

The difference between edge currents in a chiral superconductor and those in a $d + is$ superconductor can be established using a real-space lattice model on a finite cluster. For a chiral superconductor, edge currents would flow in either a clockwise or counterclockwise direction. In contrast, as we will show below, our solutions for a

finite cluster of a $d+is$ superconductor indicate that the supercurrents are not chiral and actually exhibit frustration.

We can obtain the BdG Hamiltonian in real space by performing the standard mean-field decoupling of the interaction term (8) in the pairing channel. Assuming singlet pairing only, the gap equation for order parameter $\Delta_{ij}$ for attractive interaction potential $V_{ij}$ between sites $i$ and $j$ is given by:

$$\Delta_{ij} = -V_{ij}\text{Tr}\left[\frac{\partial H}{\partial \Delta_{ij}^*}[Uf(E)U^\dagger]_{ij}^{\alpha\beta}\right] \qquad (18)$$

where the unitary operator $U$ diagonalizes the Hamiltonian $H$ such that $U^\dagger H U = E$ where $E$ is a diagonal matrix of eigenvalues and $\alpha,\beta$ are BdG degrees of freedom. The details of the self-consistent numerical calculations for the lattice are provided in Appendix A. We use Eq. (12) to calculate supercurrents on the lattice which can be recast as the following expression (derived in Appendix A), to calculate the bond currents

$$J_{ij} = -\frac{2e}{\hbar}t_{ij}\sum_\alpha \text{Im}\left[Uf(E)U^\dagger\right]_{ij}^{\alpha\alpha}. \qquad (19)$$

It is important to note that self-consistent calculations are necessary to ensure current conservation, especially in the presence of frustration. Specifically, we numerically iterate the gap equation (18) with the diagonalization of the real-space lattice BdG Hamiltonian until the gap function solution has converged to a desired accuracy. We then check that the current conservation is satisfied by computing the lattice divergence of the current on each site of the lattice.

We model a $d_{xy}$ substrate as a cluster with $N_x \times N_y$ sites and periodic boundary conditions. The $s$-wave layer is then added as a smaller island with open boundaries on top of the $d-$wave superconductor. This setup is the finite-size analog of the step-edge configuration in the long-strip geometry. In the $d$-wave layer, the periodic boundary conditions avoid any pair-breaking edges and preclude the development of any extended $s$-wave order parameter. The edges we consider are those of the $s$-wave island.

Fig. 4 shows the results of our fully self-consistent calculations of bond currents for various island geometries. In each case the island bulk stabilizes the $d_{xy}+is$ superconducting order parameter. At adjacent (100) and (010) edges of the island that form right angles with each other, supercurrents flow in opposite directions. Purely looking at the edge, it may appear that these supercurrents are converging at a point or diverging from a point, similar to the case when source or drain terms are present. However, there are no such terms present; instead, a fully self-consistent calculation ensures current conservation and the edge current frustration is resolved by compensating supercurrents flowing through the bulk.

It is useful to visualize the magnetic field patterns created by the frustrated edge and bulk currents, Fig. 4(d-f). Treating the current on each bond as a small finite-sized wire, we can calculate the magnetic field generated by it at each point on the lattice using the standard Biot-Savart law. The magnetic field (in arbitrary units) at position $\mathbf{r}$ due to a bond current flowing from lattice $i$ to lattice $j$ is

$$\mathcal{B}_{ij}(\mathbf{r}) \simeq J_{ij}\int \frac{\mathbf{dl}\times\mathbf{r}}{\mathbf{r}^3}, \qquad (20)$$

where $\mathbf{dl}$ is the line element along the bond. The net magnetic field at position $\mathbf{r}$ on the lattice is then given by $\mathbf{B}_{\text{net}}(\mathbf{r}) = \sum_{ij}\mathcal{B}_{ij}(\mathbf{r})$.

Fig. 4(d-f) illustrates the perpendicular magnetic field $\mathbf{B}_{\text{net}}(\mathbf{r})\cdot\hat{z}$ produced by the $d/s$ bilayer with islands of different shapes. We observe that different island geometries generate characteristic patterns of magnetic flux with unipolar, dipolar and quadrupolar arrangement for rectangular, triangular and square islands, respectively. These patterns are experimentally observable through local magnetometry techniques, a topic we address in more detail in Sec. V.

We conclude that the edges of the $d_{xy}+is$ superconductor always show edge current frustration, however, the manner in which the current conservation gets resolved through the bulk depends on the shape, size and symmetry of the islands.

## IV. GINZBURG-LANDAU THEORY ON THE LATTICE

We now focus our attention on the GLtheory for the $d/s$ bilayer system to explore these phenomena on longer lengthscales than what is feasible with the microscopic theory. We solve multi-component GL equations that follow from the free energy density Eq. (3) iteratively on a lattice to obtain solutions for the $d$- and $s$-wave order parameters [20, 21], such that the bulk stabilizes a $d_{xy}+is$ order parameter. We obtain solutions for $s$-wave islands in different geometries, evaluate supercurrent density in each of these systems and study edge current effects and spontaneously developed magnetic flux patterns.

### A. GL equations for the $d/s$ bilayer

We obtain the GL equation by standard variation of the free energy $F$ associated with the density (3) with respect to $s^*$ and $d^*$. The functional derivative of a generic function $f[\rho(r)]$ with respect to a parameter $\rho(r)$ is given by $\frac{\delta f}{\delta \rho} = \frac{\partial f}{\partial \rho} - \nabla\cdot\frac{\partial f}{\partial \nabla\rho}$. Working again at zero applied field we obtain a pair of partial differential equations

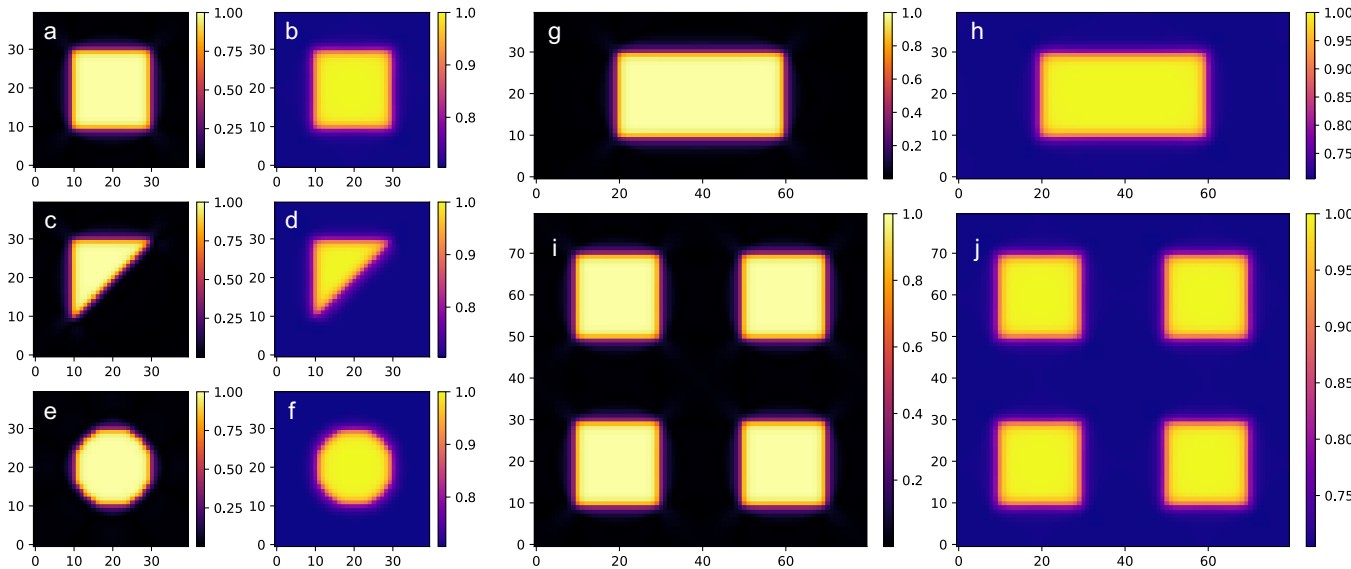

FIG. 5. Self-consistent solutions of the Ginzburg-Landau equations on a $d/s$ bilayer lattice (Eq. (24)) are presented for various shapes of $s$-wave islands with periodic boundary conditions. Panels (a) and (b) show the solutions for the $s$-wave and $d$-wave components, respectively, in the case of a square-shaped $s$-wave island. Panels (c) and (d) illustrate the solutions for a triangular $s$-wave island. Similarly, panels (e) and (f) depict the solutions for a circular $s$-wave island, while panels (g) and (h) show the solutions for a rectangular $s$-wave island. Finally, panels (i) and (j) provide the solutions for an array of $s$-wave islands, featuring four islands in total. Simulation parameters: $\alpha_d = -1$, $\alpha_s = -1$ (within the island), $\alpha_s = +1$ (outside the island), $\gamma_s = \gamma_d = 1$, $\beta_1 = \beta_2 = \beta_3 = \beta_4 = 1$.

$$-\gamma_s \nabla^2 s + \alpha_s s + 2\beta_1 |s|^2 s + \beta_3 |d|^2 s + 2\beta_4 s^* d^2 - 2\gamma_v \partial_x \partial_y d = 0,$$
$$-\gamma_d \nabla^2 d + \alpha_d d + 2\beta_2 |d|^2 d + \beta_3 |s|^2 d + 2\beta_4 d^* s^2 - 2\gamma_v \partial_x \partial_y s = 0, \tag{21}$$

which must be solved for the order parameter fields in a given geometry.

In the equations above, we take the coefficients $\alpha_s = \bar{\alpha}_s(T - T_s)$ and $\alpha_d = \bar{\alpha}_d(T - T_d)$ where $T_s$ and $T_d$ are the critical temperatures of $s$ and $d$ superconductors respectively. The coefficients $\beta_1$, $\beta_2$, $\beta_3$, $\beta_4$, $\gamma_s$ and $\gamma_d$ are all positive as suggested by lattice models. Further, we also assume $\gamma_v > 0$.

### B. Numerical solution on the lattice

We employ method described in Refs. [20, 21] to solve the above GL equations (21) discretized on a square lat-

tice. To this end we define an $N \times N$ lattice with co-ordinates $\mathbf{r} = (x, y)$. The fields $s$ and $d$ are discretized such that at each lattice point $s_r$ and $d_r$ are obtained as solutions of Eqs. (21).

To obtain the solutions of the GL equations (21), we need to define the derivative operators and the Laplacian on a lattice. The lattice derivative operators of a generic function is defined as

$$\nabla_x f(x, y) = \frac{f(x + h, y) - f(x - h, y)}{2h}, \tag{22}$$

and similar for $\nabla_y$. The Laplacian is defined as

$$\nabla^2 f(x, y) = \frac{[f(x + h, y) - 2f(x, y) + f(x - h, y)] + [f(x, y + h) - 2f(x, y) + f(x, y - h)]}{h^2}. \tag{23}$$

Defining the derivative operators this way allows us to rewrite the GL equations (21) as a coupled non-linear system of the form

$$\begin{pmatrix} -\gamma_s \nabla^2 + 2\beta_1 |s|^2 + \beta_3 |d|^2 & -2\gamma_v \nabla_x \nabla_y + 2\beta_4 d s^* \\ -2\gamma_v \nabla_x \nabla_y + 2\beta_4 s d^* & -\gamma_d \nabla^2 + 2\beta_2 |d|^2 + \beta_3 |s|^2 \end{pmatrix} \cdot \begin{pmatrix} s \\ d \end{pmatrix} = \begin{pmatrix} -\alpha_s s \\ -\alpha_d d \end{pmatrix}. \tag{24}$$

Here $s = (.., s_r, ..)^T$ and $d = (.., d_r, ..)^T$ are $N^2$-component vectors representing the discretized order parameters. If we view the matrix on the left hand side as fixed then Eq. (24) represents a set of $N^2$ coupled linear equations whose solutions correspond to the solutions of the GL equations (21) on the lattice. We start with an initial ansatz for $s$ and $d$ defined at each lattice point to construct the matrix, then solve the Eq. (24) for new $s$ and $d$. We iterate this procedure until the order parameters stop changing to obtain self-consistent solutions of GL equations.

To model an $s$-wave island deposited on a large $d$-wave substrate we vary the parameter $\alpha_s$ such that $\alpha_s < 0$ within the island and $\alpha_s > 0$ outside. We find solutions for different shapes of the island shown in Fig. 5 to study the effects of edge orientation on supercurrents. Parameters are chosen such that in the bulk the order parameter is $d_{xy} + is$ in each configuration. We consider a square, rectangular, triangular and circular $s$-wave islands. For the $d$-wave order parameter we assume periodic boundary conditions for each configuration.

## C. Currents

The full expression for the supercurrent density can be obtained, once again, by varying the free energy with respect to $\boldsymbol{A}$. One thus obtains

$$\begin{aligned} \mathbf{j} = e[d^*\{\gamma_d(\boldsymbol{\Pi}d) + \gamma_v[\hat{x}(\Pi_y s) + \hat{y}(\Pi_x s)]\} \\ + s^*\{\gamma_s(\boldsymbol{\Pi}s) + \gamma_v[\hat{x}(\Pi_y d) + \hat{y}(\Pi_x d)]\}] \\ + \text{c.c.} \end{aligned} \tag{25}$$

Here, the vector $\mathbf{j}$ is defined at each point $\mathbf{r} = (x, y)$ on the lattice.

For current conservation, we require

$$\nabla \cdot \mathbf{j} = 0 \tag{26}$$

One can show that Eq. (26) holds when the supercurrent is derived from the order parameters that satisfy the GL equations (21). However, we find that when we solve for the order parameters on a lattice, due to discretization errors, the current conservation equation (26) does not hold at points of discontinuity in the order parameters. Here, we prescribe a method to calculate the supercurrents from discrete order parameters such that the discontinuity in the order parameters does not affect current conservation on the lattice.

We know that divergence of a curl is identically zero. To enforce current conservation in our numerical solutions we therefore express the current as a curl of magnetic field $\mathbf{h}$,

$$\mathbf{j} = \nabla \times \mathbf{h}. \tag{27}$$

Taking the curl of the above equation gives $\nabla \times \mathbf{j} = -\nabla^2 \mathbf{h}$ where we used the fact that $\nabla \cdot \mathbf{h} = 0$. This gives

$$-\nabla^2 h = \nabla_x j_y - \nabla_y j_x \tag{28}$$

where $\mathbf{h}$ is taken to be along the $z$ direction. After finding the solution for $s$ and $d$ order parameters from the discretized GL equations, we calculate supercurrents on the lattice using the following steps: (i) First, we calculate the initial value of supercurrents from Eq. (25) with discretized derivative operators. (ii) Next, we find $h$ from Eq. (28) at each point on the lattice. (iii) Finally, we calculate the conserved supercurrents from Eq. (27). This procedure ensures that supercurrents remain conserved at each point on the lattice even in the presence of any discontinuity in the self-consistent solutions of the order parameters.

We calculate the supercurrents for each configuration of the $d/s$ bilayer shown in Fig. 5 and plot them, along with the corresponding magnetic fields in Fig. 6. In the microscopic model, we found the $d_{xy}$ edge of a square or a rectangular $s$-wave island to support supercurrents flowing parallel to the edge. The supercurrents from the GL theory likewise show this effect with current frustration at the corners of the $s$-wave island. To obey current conservation, the currents flow into the bulk and form a vortex-like pattern. In the triangular island configuration, the two edges forming the right angle are parallel to the $d_{xy}$ edge but the third edge is at $45°$ angle. We can think of this edge as being aligned with the $d_{x^2-y^2}$ edge, and indeed there is no current flowing parallel to it. We also observe currents flowing in opposite directions at the edges forming the right angle, returning back through the bulk of the island. On the circular island, the currents resolve their frustration by forming vortices along the edges on the outside of the island.

Finally, we also model an array of islands on a large $d$-wave substrate as shown in Fig. 6(e). Within each island, the edge currents show frustration as expected. However, to ensure current conservation a large portion of the supercurrents flow through the bulk of the $d$-wave superconductor. As a result, a vortex-antivortex lattice of sorts is formed between the islands. We remark that unlike the Abrikosov vortex lattice, the magnetic flux associated with individual vortices is not quantized here.

## V. ESTIMATION OF EDGE CURRENTS AND MAGNETIC FIELDS

The intriguing effects explored in previous sections can serve as a probe for the time-reversal-symmetry breaking $d + is$ order parameter in a $d/s$ bilayer system. We consider a substrate made of a high-$T_c$ cuprate such as Bi-2212 which is a confirmed $d$-wave superconductor and cleaves easily to obtain atomically flat surfaces. Although any $s$-wave superconductor could be used for the island, we focus on iron-based superconductors, which are known for their large superconducting gap with an $s$-wave order parameter and a high critical temperature [22]. Although the order parameter in iron-based superconductors can exhibit complex subdominant pairing channels and multiple Fermi surfaces, for the sake of simplicity we consider

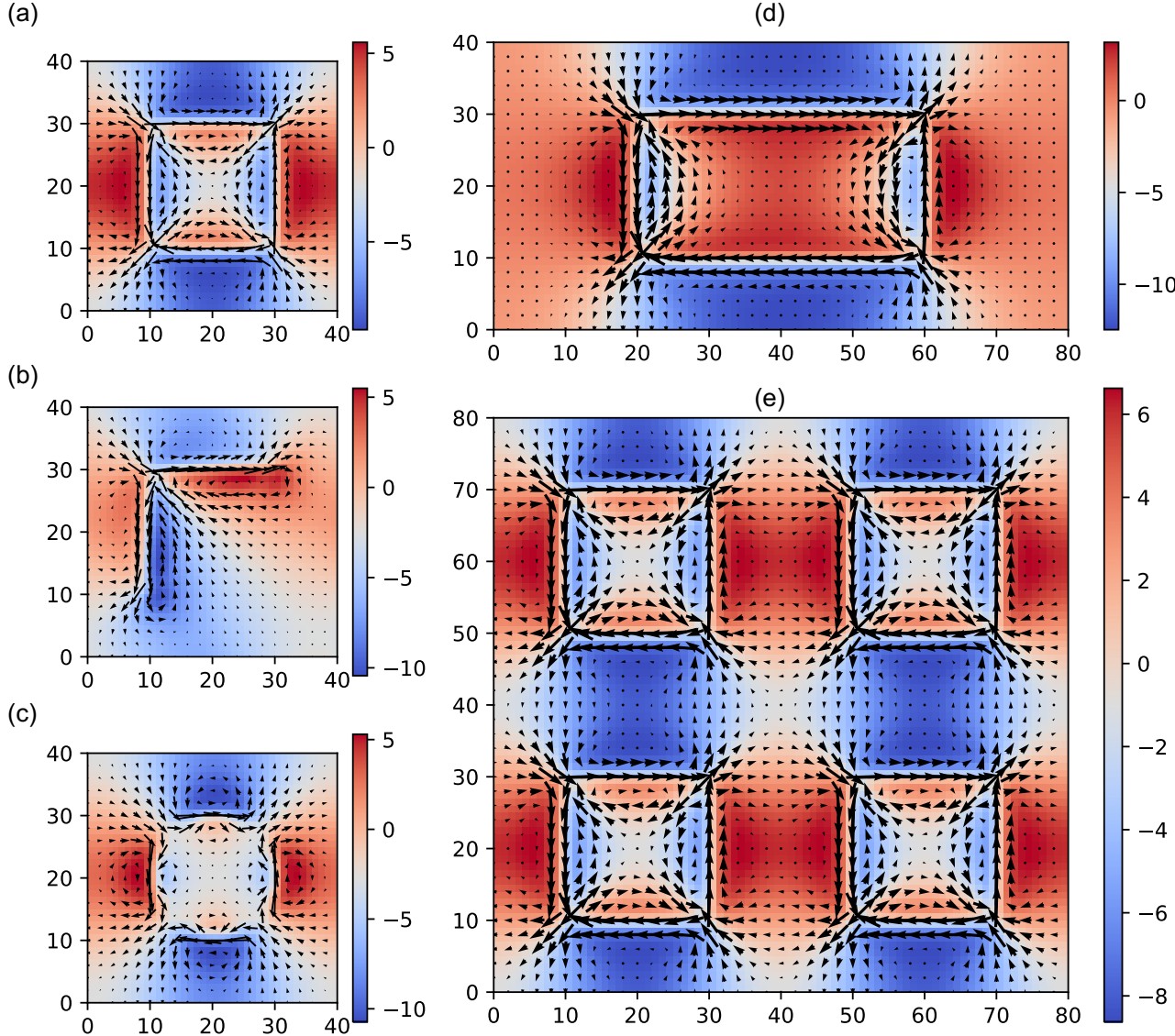

FIG. 6. Supercurrents obtained on a $d/s$ bilayer lattice obtained with the procedure prescribed in Sec. IV C using the self-consistent Ginzburg-Landau order parameters shown in Fig. 5. Panel (a) illustrates the supercurrents in a geometry featuring square-shaped $s$-wave islands. Panel (b) depicts the supercurrents flowing in a system with triangular $s$-wave islands. Panel (c) shows the supercurrents in a system with circular $s$-wave islands, while panel (d) presents the supercurrents for a system with rectangular $s$-wave islands. Finally, panel (e) displays the solutions for supercurrents in an array of $s$-wave islands (four islands featured here). Simulation parameters are same as Fig. 5.

here a minimal model with a single dominant Fermi surface and a uniform $s$-wave order parameter. We estimate the magnitude of the supercurrent along the edge of a long strip based on a fully self-consistent solution of the BdG theory, using parameters relevant to Bi-2212 and typical values for iron-based superconductors.

We show the superconducting gaps as a function of temperature in Fig. 7(a). For the $d$-wave superconducting layer, we set our normal state parameters as $t_d = 0.38$ eV, $\mu_d = -1.2t_d$ and the interaction strength as $V_d = 0.22$ eV. These parameters give us a near-circular

Fermi surface, a $d$-wave superconducting order parameter with maximal gap of approximately $\Delta_d = 50$ meV at the Fermi surface in the $d$-wave layer, in accord with the observed gap size in optimally doped Bi-2212 at low temperatures. ARPES data indicate the gap in iron-based superconductors to range from 2 meV to 30 meV [22]. In the $s$-wave layer, we choose the parameters $t_s = 0.2$ eV, $\mu_s = -1.2t_s$ and the interaction strength $V_s = 0.09$ eV. This gives us a maximal gap of $\Delta_s = 18$ meV at the Fermi surface, which is in the median range of observed ARPES gap values. The critical temperatures of the iron-

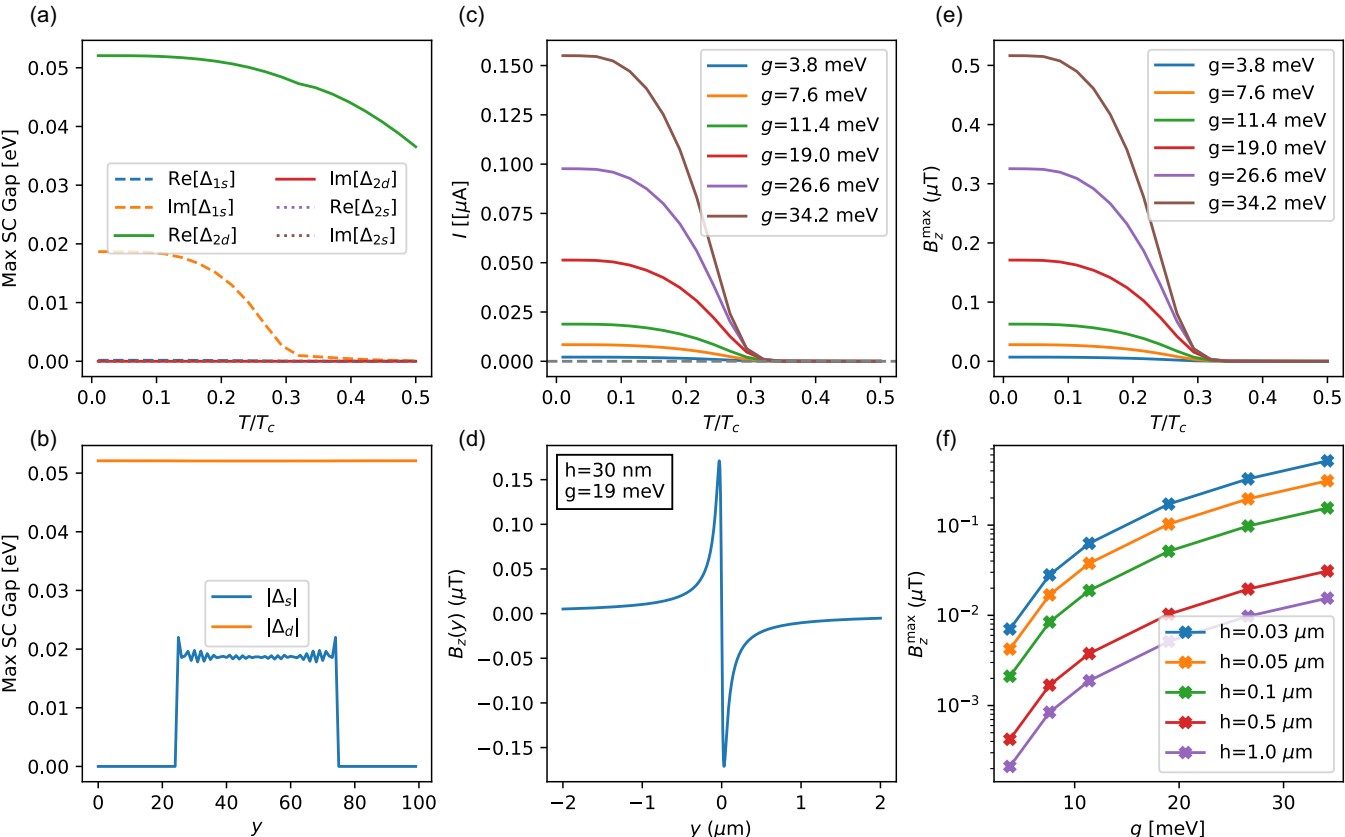

FIG. 7. Order parameters, supercurrents, and magnetic fields for d/s bilayer system comprising proximity coupled BSCCO cuprate flake ($d-$wave) with iron-based superconducting island ($s$-wave). Parameters: $t_d = 0.38$ eV, $t_s = 0.2$ eV, $\mu_d = \mu_d = -1.2t_d$. Panel (a) illustrates the maximal superconducting gap (in eV) at Fermi surface corresponding to the bulk superconducting order parameters (shown in legend) as a function of temperature for decoupled layers arranged in a long-strip geometry. Interaction strengths chosen such that the maximal gap is 0.05 eV in the $d_{xy}$ layer and 0.018 eV on the $s-$ island at low temperatures. Here, $T_c$ represents the critical temperature of the d-wave layer. Panel (b) depicts the superconducting order parameters at $T = 0.01T_c$ as a function of the strip length $y$ showing the step edges in the $s-$wave layer. Panels (c) shows the net supercurrents at a step edge as a function of temperature for inter-layer couplings ranging from 3.8 meV to 34.2 meV. Panel (d) shows the magnetic field profile of a current-carrying wire (located at [y,z]=[0,0]) as detected by SQUID loop located at a height of 30 nm. Panel (e) shows the maximum perpendicular magnetic field due to the net currents at the step edges of the system as a function of temperature and different interlayer couplings. Panel (f) shows the maximum magnetic field as a function of inter-layer coupling value for different the sensor-sample distances.

based superconductors can range from 10 K to 40 K while those in cuprates such as Bi2212 can be as high as 90 K. In our model, we obtain the critical temperature of the s-wave to be $0.4T_c$ of the d-wave layer in accordance with experimentally observed values. We assume weak inter-layer coupling values ($< 0.1t_d$) ranging between 4 meV to 30 meV.

We work with the step edge configuration on of the long strip geometry as described in Sec. III. For the model parameters specified above, we plot the order parameters at $T = 0.01T_c$. As we show in Fig. 7(b), the step edges are present in the $s$-wave layer while the $d_{xy}$ layer is uniform and exhibits no pair-breaking edges.

We compute the supercurrents based on Eq. (12) along the periodic direction $x$ as a function of $y$ for the strip geometry with finite width along $y$ for the step edge config-

uration. As there is no frustration in this geometry supercurrents are localized near the step edges. We calculate the current density $j_{\hat{x}}(y)$ and define the net supercurrent associated with a given edge as $I_{\text{net}} = \sum_{y=0}^{N_y/2} j_{\hat{x}}(y)$. In Fig. 7(c), we plot the net supercurrents as a function of temperature for different values of interlayer couplings. As the interlayer coupling decreases, the magnitude of the supercurrents also diminishes, suggesting that they stem solely from the order parameter of the bilayer rather than from any individual layer. We observe that the supercurrents generally increase as temperature decreases. At very low temperatures (below $T = 0.01T_c$ for the chosen parameters), the magnitude of the supercurrents exhibits saturation.

Edge currents in chiral superconductors produce a magnetic field that can, in principle, be detected by

state-of-the-art magnetic probes [23–25] based typically on a scanning superconducting quantum interference device (SQUID) microscope. The sensitivity and versatility of scanning SQUID have been instrumental in non-invasive measurements of a broad range of electronic orders such as those in unconventional superconductivity [26, 27], exotic magnetism [28], topological states [29], and more [30].

A typical magnetic field sensor is located at a height $h > 20$ nm above the sample. In this geometry, the edge current can be treated as a line current concentrated at the edge for all practical purposes [25]. We thus model the edge as a thin long wire along $x$ carrying the net current $I_{\text{net}}$, located at $(y, z) = (0, 0)$. The magnetic field generated by this line current at height $h$ above the sample can be deduced from Ampére's law and is given by

$$\mathbf{B} = \frac{\mu_0 I_{\text{net}}}{2\pi r} \hat{\theta} = \frac{\mu_0 I_{\text{net}}}{2\pi r} \left( \frac{y}{r} \hat{z} - \frac{h}{r} \hat{y} \right), \qquad (29)$$

where $y$ denotes the lateral position of the sensor and $r = \sqrt{y^2 + h^2}$ is the radial distance from the wire.

The flux picked up by the SQUID loop arises from the $z$−component of the magnetic field in Eq. (29). The magnetic field profile of a current-carrying wire as detected by SQUID loop located at height of 30 nm, is shown in Fig. 7(d). The net supercurrents at a step edge when treated as line current will exhibit a similar profile. We plot the peak magnetic field generated at the step edge as a function of temperature and confirm that the peak magnetic field shows similar features as the supercurrent plot in Fig. 7(e). We observe that at a height of 30 nm, the maximum magnetic field can range from $0.1 - 0.6 \ \mu\text{T}$.

Magnetic field detection depends on the vertical distance of the SQUID loop from the sample. The minimum sensor-sample distance varies based on the SQUID design, which typically aims to balance spatial resolution with magnetic field sensitivity [25, 30]. For planar SQUID, this distance is 100-500 nm [31, 32], while for SQUID-on-tip and nano-SQUID, it is 20-100 nm [29, 33]. We calculate the magnetic fields generated from the step edge of the $d+is$ bilayer for different heights. As we show in Fig. 7(f), our estimates indicate that for heights less than $0.1 \ \mu$m, the magnetic fields for all values of $g$ remain in the detectable range of the state-of-the-art SQUIDs. Further, for heights between 0.1-1$\mu$m, the magnetic field falls within the detectable range when $g > 10$ meV which is the range of the expected interlayer coupling values.

## VI. CONCLUSIONS

One of the most remarkable properties of superconductors is their ability to carry persistent currents, that is, electrical currents that cannot decay because they flow in the true ground state of the system. A classic example of this phenomenon is persistent current in a SC ring threaded by non-zero magnetic flux that serves to break

time reversal symmetry $\mathcal{T}$ and controls the current direction and magnitude. In this paper, we investigated a situation in which a heterostructure composed of a $d$-wave and an $s$-wave superconductor enables persistent current to flow in the absence of applied magnetic field. Such a heterostructure breaks $\mathcal{T}$ spontaneously and stabilizes a $d + is$ or a $d - is$ order parameter in the bulk. Remarkably, for some geometries, this spontaneous breaking of time-reversal symmetry manifests itself in the form of edge currents.

While chiral edge currents are known to occur in superconductors with non-trivial topology, it is important to note that a $d \pm is$ superconductor is topologically trivial. As well, the edge currents in this system are not chiral but give rise to various intriguing effects, which we explore and explain in this work. Specifically, unlike chiral edge currents which flow around the perimeter of the sample, we find that $d \pm is$ superconductors exhibit *frustrated* edge currents that appear to emanate from and disappear into sample corners.

This stark difference between the edge currents of a chiral $d$-wave superconductor and a $d \pm is$ superconductor becomes evident when we solve relevant microscopic models and Ginzburg-Landau theory for $s$-wave islands in various geometries placed on top of a $d$-wave substrate. We find that any two perpendicular edges aligned with nodal directions of the $d$ order parameter host supercurrents that flow in *opposite* directions as indicated in Fig. 1(b), giving the appearance of the supercurrents either flowing into or out of a corner. However, since there are no source terms present, the current must be strictly conserved. Fully self-consistent calculations that ensure current conservation show that this frustration is resolved by current flowing through the bulk of the system, creating vortex-like patterns of superflow. While these are not flux-quantized vortices, they nevertheless generate characteristic magnetic field profiles that can be used to experimentally detect these phenomena.

To observe these effects we proposed a specific material system consisting of a BSCCO substrate with a $d$-wave order parameter and an $s$-wave iron-based superconductor forming an island. Breaking of the time-reversal symmetry in this system will realize the $d \pm is$ superconducting phase with frustrated edge currents of the type discussed above. We showed that, for a wide range of model parameters, the magnetic fields resulting from the resulting vortex-like superflow patterns are strong enough to be detected by state-of-the-art scanning SQUID microscopes.

More broadly, the phenomenon of the frustrated edge currents explored in this work touches on many key concepts in modern condensed matter physics. Frustration is central to the field of quantum magnetism where it underlies such phenomena as exotic phase transitions and the emergence of the elusive spin-liquid phases [34]. Using the $d/s$ bilayer platform with a periodic array of nanoscale islands one could imagine engineering specific patterns of edge currents that would realize the types

of frustration required in models of classical or quantum spin liquids. Quantum fluctuations in such islands arise naturally when one includes the effect of charging energy, as discussed recently in the context of an improved transmon qubit based on a $d/s$ bilayer [15], and can be precisely tailored by adjusting the island size and geometry. Another promising avenue of research would be to explore the role of topology in this system. Some iron-based superconductors are known to host topological surface states along with a bulk $s-$wave order parameter [35–42]. Quantized vortices that can potentially host Majorana particles have been observed in such systems. If a cuprate $d$-wave flake is proximity coupled to such a system, it would be interesting to study the interplay of the supercurrent effects arising purely due to the time-reversal symmetry breaking of a $d \pm is$ system and those due to the presence of topological surface states.

## VII. ACKNOWLEDGMENTS

We thank Catherine Kallin, Thomas Scaffidi, Alberto Nocera, Niclas Heinsdorff and Nitin Kaushal for helpful discussions and correspondence. This work was supported by NSERC, CIFAR and the Canada First Research Excellence Fund, Quantum Materials and Future Technologies Program.

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

## Appendix A: Self-consistent calculations in the BdG model

### 1. Self-consistent treatment on a lattice and the supercurrent

Assuming singlet pairing only, the gap equation for order parameter $\Delta_{ij}$ for attractive interaction potential $V_{ij}$ between sites $i$ and $j$ is given by:

$$\Delta_{ij} = -V_{ij}\text{Tr}\left[\frac{\partial h}{\partial \Delta_{ij}^*}\frac{1}{\beta}\sum_{\omega_n}\mathcal{G}(i\omega_n)\right] \quad (A1)$$

which can be derived as the saddle point of the path integral, as described in [7]. We define the unitary operator $U$ that diagonalizes the hamiltonian $H$ such that $U^\dagger H U = E$ where $E$ is a diagonal matrix of eigenvalues. Using the definition of the Green's function $\frac{1}{i\omega_n - E} = U^\dagger \mathcal{G}(i\omega_n)U$ we obtain

$$\frac{1}{\beta}\sum_n U^\dagger \mathcal{G}(i\omega_n)U e^{i\omega_n 0^+} = \frac{1}{\beta}\sum_n \frac{e^{i\omega_n 0^+}}{i\omega_n - E} = f(E) \quad (A2)$$

where multiplied the equation by convergence factor and summed over all fermionic Matsubara frequencies. Note that the RHS of this equation is the fermi factor which we denote $f(E)$ as a diagonal matrix, similar to $E$ we defined above. Inverting this expression we find that

$$\frac{1}{\beta}\sum_n \mathcal{G}_{ij}^{\alpha\beta}(i\omega_n)e^{i\omega_n 0^+} = [Uf(E)U^\dagger]_{ij}^{\alpha\beta} \quad (A3)$$

### 2. Current operator

Using the Heisenberg equation of motion $dN/dt = i/\hbar[H, N]$ we compute the charge flow from the degree of freedom $\mu = (r, \alpha, \sigma)$ where we packed position, orbital and spin indices together. $N = c_\mu^\dagger c_\mu$ And the net charge flow for $\mu$ us given by

$$\frac{dQ}{dt} = e\frac{dN}{dt} = \frac{ie}{\hbar}[H, N]$$

with carriers with charge $e$ for the Hamiltonian

$$H = t_{\mu\nu}c_\mu^\dagger c_\nu + H_{int}. \quad (A4)$$

The quartic interaction terms commute with $H$ since they conserve particle number on a given site. Note that we derive this operator before we do the MF decoupling since pairing terms in the SC Hamiltonian will generate additional terms. Only the hopping terms in $H$ will contribute to the current:

$$\frac{dQ_\mu}{dt} = -\frac{ie}{\hbar}\sum_\nu\left[t_{\mu\nu}c_\mu^\dagger c_\nu - t_{\nu\mu}c_\nu^\dagger c_\mu\right] \quad (A5)$$

where the first term corresponds to charge transport from orbital $\nu$ to $\mu$ and the second term corresponds to the reverse process. This we can consider as the bond current $I_{\mu\nu}$ from site $\nu$ to $\mu$:

$$I_{\mu\nu} = -\frac{ie}{\hbar}\left[h_{\mu\nu}c_\mu^\dagger c_\nu - h_{\nu\mu}c_\nu^\dagger c_\mu\right] \quad (A6)$$

More specifically we can write the current operator as

$$J_{ij} = \frac{ie}{\hbar}t_{ij}\left(c_{i\uparrow}^\dagger c_{j\uparrow} + c_{i\downarrow}^\dagger c_{j\downarrow}\right) + h.c. \quad (A7)$$

this can be rewritten in the following form which is useful to cast our equations in particle-hole basis.

$$J_{ij} = \frac{ie}{\hbar}\left(t_{ij}c_{i\uparrow}^\dagger c_{j\uparrow} + t_{ij}^*c_{i\downarrow}c_{j\downarrow}^\dagger\right) + h.c. \quad (A8)$$

Assuming $t_{ij}$ is real (and therefore symmetric), we can write the current operator defining the spinors $\Psi_i = (c_{i\uparrow}, c_{i\downarrow}^\dagger)$ which in turn becomes

$$J_{ij} = \frac{ie}{\hbar}t_{ij}\Psi_i^\dagger\begin{pmatrix}1 & 0\\ 0 & 1\end{pmatrix}\Psi_j + h.c. \quad (A9)$$

The expectation value of the current can be written

$$\begin{aligned}\langle J_{ij}\rangle &= \frac{ie}{\hbar}t_{ij}\text{Tr}\left[\Psi_i^\dagger\begin{pmatrix}1 & 0\\ 0 & 1\end{pmatrix}\Psi_j\right] + h.c.\\ &= \frac{ie}{\hbar}t_{ij}\text{Tr}\left[\Psi_j\Psi_i^\dagger\right] + h.c.\\ &= \frac{ie}{\hbar}t_{ij}\sum_\alpha\left[\langle\Psi_{j\alpha}\Psi_{i\alpha}^\dagger\rangle - \langle\Psi_{i\alpha}\Psi_{j\alpha}^\dagger\rangle\right]\end{aligned}$$

where we used the cyclic property of trace and rewrote it as sum over $\alpha$ which denotes the BdG degrees of freedom. Note that the expectation values we are interested in can be expressed as imaginary time Gorkov Green's functions

$$\mathcal{G}_{ij}^{\alpha\beta}(0^+) = -\langle\Psi_{i\alpha}\Psi_{j\beta}^\dagger\rangle \quad (A10)$$

Using the imaginary time relation $\mathcal{G}(0^-) - \mathcal{G}(0^+) = 1$ [43] and the Matsubara frequency space representation $\mathcal{G}(\tau) = \beta^{-1}\sum_n \mathcal{G}(i\omega_n)e^{-i\omega_n\tau}$ we can show that the expectation value of the bond current is given by

$$\langle J_{ij}\rangle = \frac{ie}{\hbar}t_{ij}\frac{1}{\beta}\sum_{n,\alpha}\left[\mathcal{G}_{ij}^{\alpha\alpha}(i\omega_n)e^{i\omega_n 0^+} - \mathcal{G}_{ji}^{\alpha\alpha}(i\omega_n)e^{i\omega_n 0^+}\right] \quad (A11)$$

Note that this expression appears in A11 and when combined we obtain the equation below that we use in our numerical calculations of current

$$\begin{aligned}J_{ij} &= \frac{ie}{\hbar}t_{ij}\sum_\alpha\left[Uf(E)U^\dagger\right]_{ij}^{\alpha\alpha} - \left[Uf(E)U^\dagger\right]_{ji}^{\alpha\alpha}\\ &= -\frac{2e}{\hbar}t_{ij}\sum_\alpha\text{Im}\left[Uf(E)U^\dagger\right]_{ij}^{\alpha\alpha} \quad (A12)\end{aligned}$$

This expression also appears in the gap equation 18 and simplifies it into a form that is suitable for numerical calculations.