# Peer review of "Frustrated edge currents in bilayers formed of s- and d-wave superconductors"

_SciPost Physics_

## Round 1 · Referee Report · Julien Garaud (Referee 1) · 2025-8-20

Strengths

1 - Clear motivation and presentation: The introduction is well written, places the work in context of superconducting heterostructures, and highlights potential applications.

2- Comprehensive an complementary methodology : The authors use both mean field Ginzburg–Landau (GL) and fully self-consistent Bogoliubov–de Gennes (BdG) simulations.

3- Potential experimental relevance: The estimate of current magnitudes and magnetic field signals, makes it potentially observable via SQUID microscopy.

Weaknesses

1- GL numerics neglect screening . Omitting the self-consistent coupling to magnetic fields,may qualitatively alter the patterns observed.

2- Apparent mismatch between GL and BdG – Figures (e.g. 4d vs. 6a, triangular and rectangular islands) show discrepancies in current patterns, suggesting that the GL approximation as implemented is missing key physics.

Report

The current manuscript presents an investigation of the properties of edge currents in heterostructures formed by a conventional $s$-wave superconductor and a $d$-wave superconductor ( chosen to be a high-$T_c$ cuprate). They show that it can result in an emergent $d\pm is$ order parameter that spontaneously breaks the time-reversal symmetry, and generates spontaneous currents that appear to originate or terminate at sample corners. Using both Ginzburg-Landau (GL) theory and self-consistent Bogoliubov-de Gennes (BdG) lattice simulations, they demonstrate how current conservation is restored through bulk currents and vortex-like magnetic flux patterns. Afterward they estimate magnitudes of currents and fields relevant for experimental detection with SQUIDs.

Overall I think the paper is well written and that motivations are clear. It propose a complete analysis of superconducting heterostructures with potentially interesting applications. I have a several remarks/questions, essentially of technical order.

Remarks of primary concern:

1- Numerical simulation of Ginzburg-Landau equations: I think here is a (potentially) major issue. In section IV, the authors discuss the numerical treatment of the Ginzburg-Landau equations. When deriving Eqs. (21) and (24), they explicitly set ${\bf A}=0$, and then enforce local conservation $\bf j = \nabla \times h$ afterwards. It is not equivalent to solving self-consistently the full GL equations coupled to the magnetic field $\bf B = \nabla\times A$. By this procedure of effectively decoupling the gauge field from the order parameters, the authors neglect the back reaction of generated magnetic fields on the order parameters. In other words Meissner screening is neglected, and this could be a primary source of errors on the GL part. For example the observed vortex-like patterns are not guaranteed to be the same once screening is included.

2- If I understand correctly the Fig. 6, the background colormap gives the induced magnetic field. The induced field seems quite substantial in magnitude, thus I naively expect that its back reaction would be strong.

3- When comparing the microscopic BdG calculations with the GL, for example the square Fig. 4d anf Fig. 6a. Looking on the diagonals, the current inverts its direction on the corner for GL, while it does not for BdG. Similarly also for the triangle. And it seems even worth for the rectangle where the outter field varies a lot for the GL while it does not for BdG

4- Unconventional models of superconductivity feature surface terms (see Table X of Sigrist and Ueda Rev. Mod. Phys. 63, 239, 1991). Shouldn't such terms also appear at the boundary of the s-wave flake for the coexisting $d+is$?

Remarks of lesser concern:

5- The authors use a center difference stencil that gives a second order accuracy . It is not really clear to me that this is enough to correctly address sharp discontinuities. Yet the setup seems to washout discontinuity of the gap thus making s-wave leak out the true s-wave flake. Is it correct? Also I don't recall reading about the number of lattice site.

6- Convergence criteria of iterations should be made more explicit. The authors say that they iterate until order parameters "stop changing". But they do not state tolerance or max iterations. This could be of concern since nonlinear GL systems can show slow or even oscillatory convergence.

7- In equation (28), it is not clear what are the boundary conditions for $\bf h$.

8- The uthors should clarify the concept of frustration. The paper makes the idea intuitive with corner current diagrams, but it could help to more explicitly contrast with true current sources/sinks to avoid confusions.

9- The sign of the higher order Josephson coupling ($C$ in Eq.(1-2) or $\beta_4$ for GL) is crucial for time-reversal symmetry breaking, and hence spontaneous currents. Depending on the sign it can result a $d+is$ state that breaks the time-reversal symmetry, or in a $d+s$ state that does not. In section V, the authors try to relate with experimental situation of a $s$-wave flake on Bi-2212 substrate. Is there any argument for why there would be such a coupling, instead of one that preserves the time-reversal symmetry?

The manuscript presents some interesting physics that surely deserves publication. However, before considering acceptance, I think the authors should clarify the points I raised above. Especially on the numerical treatment of the GL part and agreement with BdG.

I think that the procedure for GL neglects Meissner screening, and is thus the origin of mismatch between BdG and GL. My naive expectation is that a true self-consistent solution of the GL side would solve this major issue.

Requested changes

1- Address the questions of main concern.

Recommendation

Ask for major revision

---

## Round 1 · Referee Report · Philip Brydon (Referee 2) · 2025-9-22

Strengths

  1. The authors have proposed a plausible geometry for realising an exotic TRSB state with edge currents

  2. The authors have carefully considered the magnitude of the experimental signatures of their proposed TRSB state

  3. The authors' conclusions are bolstered by using two distinct methods

Weaknesses

  1. Insufficient acknowledgement of prior literature

  2. Insufficient technical detail

  3. Lack of critical evaluation/comparison of different methodologies

  4. Lack of discussion of confounding effects

Report

The reviewed manuscript "Frustrated edge currents in bilayers formed of s- and d-wave superconductors" by Pathak et al. presents a theoretical study of s-wave superconducting islands which are deposited on d-wave superconductors. The authors propose that the two superconducting states coexist in a time-reversal symmetry-breaking state, with a $\pm \pi/2$ phase difference between them. A consequence of this is the existence of spontaneous currents near the edge of the s-wave island, which the authors investigate using both microscopic Bogoliubov-de Gennes (BdG) models and a Ginzburg-Landau (GL) free energy. Unlike in chiral superconductors, the edge currents are not topologically protected, and indeed display "frustration" effects where currents on adjacent edges seems to flow toward or away from the shared corner, but the author's methods ensure that current conservation is satisfied. The authors claim that the magnetic fields associated with the edge currents can plausibly be detected with current apparatus sensitivities.

Spontaneous TRSB in superconducting systems is a topic of perenial fascination. The ability to engineer exotic superconducting states in heterostructures is a long-standing and compelling idea, as the conditions which favour TRSB can in principle be more carefully controlled, although in practice this has been hard to achieve or convincingly demonstrate. Recent breakthroughs in twisted bilayer cuprates has reignited interest in the field, and the reviewed manuscript is inspired by these geometries. The manuscript will certainly be of interest to the community and may help guide experimentalists; the discussion of whether or not the effect could be measurable is particularly to be applauded.

I support publication of the article in SciPost Physics, but there are a number of issues that I would first like to see convincingly addressed by the authors:

  1. The article is somewhat deficient in its citations and review of previous contributions. For example:

a) Spontaneous TRSB at the interface of a d-wave superconductor and an s-wave superconductor has been discussed for many years, see in particular

S. Yip, "Josephson current-phase relationships with unconventional superconductors", Physical Review B 52, 3087 (1995).

Edge currents in d+is islands have been self-consistently calculated in

M. H. S. Amin, "d+is versus d+id' time reversal symmetry breaking states in finite size systems", Physical Review B 66, 174515 (2002).

A more detailed review is provided by

M. Sigrist, "Time-Reversal Symmetry Breaking States in High-Temperature Superconductors", Progress of Theoretical Physics 99, 899 (1998)

b) The GL free energy studied by the authors was first proposed by

R. Joynt, "Upward curvature of $H_{c2}$ in high-$T_c $superconductors: Possible evidence for s-d pairing", Physical Review B 41, 4271 (1990)

The authors should more carefully examine the (very large) extant literature on the subject of d-s heterostructures to ensure that they present an appropriate hisotrical perspective.

  1. Despite the long study of d-s heterostructures, I am not aware of any experimental reports of a spontaneous TRSB state. Although grain boundaries within the cuprate crystal could be modelled as a d-s junction, I am thinking more of the c-axis experiments, relevant to the model studied by the authors, see e.g.

R. Kleiner et al., "Pair Tunneling from c-Axis YBa$_2$⁢Cu$_3$⁢O$_{7−x}$ to Pb: Evidence for s-Wave Component from Microwave Induced Steps", Physical Review Letters 76, 2161 (1996)

This suggests that the d+is state proposed by the authors might not emerge in practice as easily as it does in theory. The authors should certainly mention this in their manuscript.

One might speculate that surface roughness could be a factor in disrupting the TRSB state, as the scattering between different interface momenta would suppress the frustration which leads to the TRSB state. Like many such proposals, the authors have not accounted for interface roughness in their model, which assumes a uniform coupling between the s- and d-wave superconductors. I do not expect the authors to perform calculations accounting for such disorder, but it should be discussed in the manuscript.

An alternative suggestion is proposed in

W. Zhang and Z. D. Wang, "Interface roughness and proximity effect on a c-axis Josephson junction between s-wave and d-wave superconductors", Physical Review B 65, 144527 (2002)

where the authors suggest that a strong repulsive interaction for s-wave pairing in the d-wave superconductor could play a significant role in modifying the interface physics. This would seem a natural thing to include in the BdG method.

  1. The discussion of the numerical solution of the GL equations is somewhat deficient:

(a) The parameters $\beta_1$, $\beta_2$, etc. in Eq. 21 are not introduced in Eq. 3, as indicated in the text. These should be presented in order to clearly understand the results.

(b) In the numerical analysis of the GL free energy, the authors introduce the lattice spacing "h" (see e.g. Eqs. 22 and 23). The authors should explain how they choose this length scale and state its value in terms of the coherence lengths of the s- and d-wave superconductors.

Are the axis labels in figs. 5 and 6 expressed in terms of h? It would be desirable to express this in terms of a physical quantity, e.g. the correlation length of the d-wave superconductor.

  1. Comparing figures 4 and 6, one notices significant differences in the magnetic field predicted by the BdG and GL theories, e.g. the field inside and outside of the square island is oriented in the same direction in the former, but in opposite directions in the latter. The authors should comment upon this difference.

Moreover, there appears to be significant currents along the long edge of the triangle in the BdG approach but not in the GL - why is that? Since the long edge of the triangle should be $d_{x^2-y^2}$-like, the result of the slab calculations implies that we should not see any currents here. Why doesn't the BdG approach capture this?

Another surprising feature of the GL results is that the d-wave order parameters is significantly enhanced underneath the s-wave island (enhanced by about 50% of its bulk value!). Is this consistent with the BdG results? Or is it an artifact of choosing that $\beta_3=\beta_4$ in Eq. 21? Can the authors justify this choice on microscopic grounds, e.g. a microscopic calculation of the GL expansion coefficients from the BdG Hamiltonian?

  1. It appears that neither the BdG nor the GL approaches are electromagnetically self-consistent, i.e. screening currents are not taken into account. Although this is reasonable in the BdG approach, where the different length scales make this impractical, this is not necessarily true for the GL theory. For BSCCO the penetration depth is about 210nm, whereas the coherence length is 1.6nm. Assuming that the axis labels in fig. 5 are expressed in units of the coherence length, it would seem possible for the authors to examine such an effect (although admittedly numerically costly). I note that screening is accounted for in the numerical results shown by Sigrist (PTP, 1998), although the penetration depth is comparable to the coherence length in these results. In any case, the authors should comment on this in their discussion.

Requested changes

  1. The authors should give a more complete review of interface TRSB in d- and s-wave junctions. At a minimum they should cite the suggested articles.

  2. The authors should discuss confounding effects, e.g. interface disorder. The authors may also like to check the effect of a strong repulsive s-wave interaction in the d-wave superconductor within the BdG approach.

  3. The discussion of the numerical solution of the GL equations should be improved as suggested in the report. I recommend that the authors check the robustness of their results under changes to the GL free energy parameters.

  4. The authors should clearly note (and if possible, explain) the differences between the GL and BdG theory predictions.

  5. The authors should consider investigating the effect of electromagnetic self-consistency within the GL method.

Recommendation

Ask for major revision

---

## Editorial Decision

awaiting_resubmission